# Forecasting COVID-19-Associated Hospitalizations under Different Levels of Social Distancing in Lombardy and Emilia-Romagna, Northern Italy: Results from an Extended SEIR Compartmental Model

**DOI:** 10.3390/jcm9051492

**Published:** 2020-05-15

**Authors:** Chiara Reno, Jacopo Lenzi, Antonio Navarra, Eleonora Barelli, Davide Gori, Alessandro Lanza, Riccardo Valentini, Biao Tang, Maria Pia Fantini

**Affiliations:** 1Department of Biomedical and Neuromotor Sciences, Alma Mater Studiorum—University of Bologna, 40126 Bologna, Italy; chiara.reno@studio.unibo.it (C.R.); davide.gori4@unibo.it (D.G.); mariapia.fantini@unibo.it (M.P.F.); 2Euro-Mediterranean Center on Climate Change, 40127 Bologna, Italy; antonio.navarra@cmcc.it (A.N.); alessandrolanza.al@gmail.com (A.L.); rik@unitus.it (R.V.); 3Department of Physics and Astronomy, Alma Mater Studiorum—University of Bologna, 40126 Bologna, Italy; eleonora.barelli2@unibo.it; 4Department of Political Science, LUISS—Libera Università Internazionale degli Studi Sociali Guido Carli, 00197 Rome, Italy; 5Department of Innovation in Biological, Agro-Food and Forest Systems, Tuscia University, 01100 Viterbo, Italy; 6Laboratory for Industrial and Applied Mathematics, Department of Mathematics and Statistics, York University, Toronto, ON M3J 1P3, Canada; btang66@yorku.ca; 7The Interdisciplinary Research Center for Mathematics and Life Sciences, Xi’an Jiaotong University, Xi’an 710049, China

**Keywords:** coronavirus, COVID-19, SARS-CoV-2, pandemic, public health intervention, lockdown, resurgence, forecasting, mathematical modelling, SEIR model

## Abstract

The outbreak of coronavirus disease 2019 (COVID-19) was identified in Wuhan, China, in December 2019. As of 17 April 2020, more than 2 million cases of COVID-19 have been reported worldwide. Northern Italy is one of the world’s centers of active coronavirus cases. In this study, we predicted the spread of COVID-19 and its burden on hospital care under different conditions of social distancing in Lombardy and Emilia-Romagna, the two regions of Italy most affected by the epidemic. To do this, we used a Susceptible-Exposed-Infectious-Recovered (SEIR) deterministic model, which encompasses compartments relevant to public health interventions such as quarantine. A new compartment L was added to the model for isolated infected population, i.e., individuals tested positives that do not need hospital care. We found that in Lombardy restrictive containment measures should be prolonged at least until early July to avoid a resurgence of hospitalizations; on the other hand, in Emilia-Romagna the number of hospitalized cases could be kept under a reasonable amount with a higher contact rate. Our results suggest that territory-specific forecasts under different scenarios are crucial to enhance or take new containment measures during the epidemic.

## 1. Introduction

On 11 March 2020, the World Health Organization (WHO) declared the Coronavirus Disease 2019 (COVID-19) a pandemic [1]. This viral infection commonly presents with fever and cough and frequently leads to lower respiratory tract disease, with poor clinical outcomes associated with older age and underlying health conditions [2]. Broken out in China, subsequently spread in Thailand, Japan, and South Korea [3], the epidemic eventually reached Italy, which became the first European country to be affected. On 20 February 2020, a man in his 30s without a history of possible exposure abroad was admitted to an intensive care unit (ICU) of Lombardy (northern Italy) and tested positive for the virus that causes coronavirus disease 2019 [4,5]. Since then, the number of cases has quickly increased, with all Italian regions reporting patients affected with COVID-19 and with a marked involvement of northern Italy [5]. To date, Lombardy and Emilia-Romagna report the highest numbers in terms of both cases and deaths [6]. Several different measures have been put in place, from the closure of the schools and museums to the closure of restaurants, commercial activities, and in general of public activities implying crowding of people, until the total lockdown of the country, which was declared on 9 March 2020 [7].

The COVID-19 outbreak had a severe impact on the Italian health services, which had to face and adjust to the rapidly changing situation. Because of the great share of cases that need to be hospitalized and of the prolonged hospital stay [6,8], the adaptation of the hospital capacity was a critical issue, with particular regard to ICUs. In Lombardy, from day 1 to day 14 of the emergency, there has been a steep and steady increase in ICU admissions; on March 7, the total number of patients with COVID-19 admitted to ICUs represented 16% of the total hospitalized patients with COVID-19 [4]. Over the first 18 days, the COVID-19 Lombardy ICU network, set up to face the emergency, created 482 ICU beds ready for patients [4]. In the following days, the Italian regions planned to add and create new ICU beds relying on forecasts of estimated ICU demand [9].

On 13 April 2020, there were 12 028 hospitalized symptomatic patients in Lombardy, the highest number in Italy, while Emilia-Romagna had the second-highest number, 3490 (altogether, 55% of all hospitalized patients in Italy [6]).

As the epidemic rapidly spreads, the daily updated great amount of data can be examined using different types of methods, allowing researchers to closely investigate the course of the pandemic.

In particular, the emerging and re-emerging of infectious diseases have led over the years to the development of mathematical models that have become significant tools to analyze the virus spreads, thus contributing to the planning and improvement of strategies to control the transmission of these diseases [10]. Governments across the world rely on projections provided by mathematical models to make crucial decisions during this pandemic.

There are different approaches to model a complex phenomenon like the outbreak of a new infectious disease: the equation- and the agent-based are the most adopted [11]. In this study, we use an equation-based model, but provide features of both these approaches to highlight the main conceptual traits of the model we decided to adopt here.

The equation-based approach, also known as meta-population, characterizes models where each individual is not tracked throughout the model; rather, the population under observation is divided into compartments and their evolution is modelled mainly through differential equations. The simplest example of a meta-population model is the SEIR (Susceptible-Exposed-Infectious-Recovered) model, developed by Kermack and McKendrick [12]. References for the meta-population approach include [13,14,15]. On the opposite, the agent-based approach consists in modelling individual characteristics as well as movements and contact patterns of individual people, named agents, within a population under study. References for the agent-based approach to epidemic modelling include [16,17,18,19].

Some authors have been comparing the effectiveness of agent-based and equation-based models for infectious disease epidemiology. On one side, the agent-based approach allows to capture very naturally the heterogeneity of the population to give a more precise view of an outbreak and its evolution [20]. On the other side, meta-population models provide less detailed information than their agent-based counterparts but are fairly scalable and can provide scenarios with thousands of stochastic runs [21].

As the epidemic of the novel coronavirus broke out, an increasing number of models have been published, and are becoming more and more refined as the knowledge on the disease progresses. One of the first models to be proposed is an equation-based one [22,23], focused on the estimation of the transmission risk of COVID-19, its impact on health services capacity and its implications for public health interventions. This model is of relevance, referring to the first big outbreak occurred in China, from where the epidemic subsequently spread all over the world.

The aim of our study was to implement the model by Tang et al. and adapt it to the Italian context, and to forecast the spread of the infection and its burden on hospitalizations under different conditions of social distancing in Lombardy and Emilia-Romagna, the two regions of Italy most affected by the epidemic. This is of particular interest when it is necessary to rapidly adapt the hospital and services organization and make decisions on containment measures.

## 2. Experimental Section

### 2.1. Model Specification

We used a SEIR deterministic epidemiological model, which encompasses compartments relevant to public health interventions such as quarantine and isolation. As shown in Figure 1, we added to a SEIR-based model estimated on Chinese data [22] a new compartment L for isolated infected population, i.e., individuals tested positives that do not need hospital care. Hereinafter, we will refer to this model as extended SEIR.

The other compartments were those proposed by Tang and colleagues [22]: susceptible (S), exposed (E), infectious but not symptomatic (A), infectious with symptoms (I), hospitalized (H), recovered (R), quarantined susceptible ($S_{q}$), and quarantined exposed ($E_{q}$) populations. However, in Tang et al. [22], the hospitalized are all individuals intercepted by health services as sick, while in Italy’s organizational system the cases recorded may either end up in the hospital (H) or stay at home if the symptoms are mild and the housing conditions are adequate (L). Therefore, we have introduced a new compartment (*L*) to take into account also this sector (70% of all active cases in Italy as of April 13 [6]). The model introduced in Tang et al. [22] was then modified by including the following equations to handle the transmission dynamics from and to L:$$S^{\prime }=-\left(\beta c+cq\left(1-\beta \right)\right)S\left(I+\theta A\right)+\lambda S_{q},$$
$$E^{\prime }=\beta c\left(1-q\right)S\left(1+\theta A\right)-\sigma E,$$
$$I^{\prime }=\sigma \rho E-\left(\delta_{I}+\varepsilon_{I}+\gamma_{I}+\alpha \right)I,$$
$$A^{\prime }=\sigma \left(1-\varrho \right)E-\gamma_{A}A,$$
$$S_{q}^{'}=\left(1-\beta \right)cqS\left(I+\theta A\right)-\lambda S_{q},$$
$$E_{q}^{'}=\beta cqS\left(I+\theta A\right)-\left(\delta_{q}+\varepsilon_{q}\right)E_{q},$$
$$L^{\prime }=\varepsilon_{q}E_{q}+\varepsilon_{I}I-\left(\delta_{L}+\gamma_{L}+\alpha \right)L,$$
$$H^{\prime }=\delta_{I}I+\delta_{q}E_{q}+\delta_{L}L-\left(\gamma_{H}+\alpha \right)H,$$
$$R^{\prime }=\gamma_{I}I+\gamma_{A}A+\gamma_{H}H+\gamma_{L}L,$$
where $\varepsilon_{q}$ is the home isolation rate for quarantined exposed, $\varepsilon_{I}$ is the home isolation rate for non-quarantined infected, $\delta_{L}$ is the hospitalization rate for isolated infected that we assume equal to 20%, and $\gamma_{L}$ is the recovery rate for isolated infected individuals. Assuming that one in four tests positive and that 80% of the positives do not need acute hospital care [6], we estimated that $\varepsilon_{q}=\varepsilon_{I}=0.20$; we also assumed $\gamma_{L}$ to be equal to the recovery rate for asymptomatic individuals $\gamma_{A}$ (0.14). All the other parameters were initialized with the values proposed by Tang et al. [22], with the exception of the infection rate for asymptomatic individuals (θ), which was assumed to be 0.05 [24] as opposed to 0 (Table 1). In Section 2.2 we provide details about c and $\delta_{I}$, which are respectively the contact function and the diagnosis rate function, both of time variable t.

### 2.2. Formulation of the Model

The model is essentially a coupled system of nonlinear ordinary differential equations that produce the evolution of the compartments over time. The model was initialized with the regional data of Lombardy and Emilia-Romagna released by the Civil Protection Department of Italy on 9 March 2020 (the first day of the national quarantine). Other parameters we entered, such as population sizes on day 0 (9 March) have been obtained by gathering information from other official statistics or making some assumptions for inputting missing data. More specifically, we assumed that the ratio of undetected to detected positive cases was 10 to 1, and that asymptomatic individuals constituted 66% of the infected pool. To check the robustness of our forecasts, we performed some sensitivity analyses by making different assumptions for these compartments. In particular, we assumed that the proportion of undetected infected individuals was 83% (i.e., 5 to 1 as opposed to 10 to 1 (91%)) and that the proportion of asymptomatic infections was one-third instead of two-thirds. No alternative assumptions were made in relation to number of undetected exposed individuals and to the case mix of quarantined individuals, given the lack of available information in the literature.

The initial values are shown in Table 2.

The containment measures have been parameterized via the “contacts” function $c\left(t\right)$. We have assumed that the implementation of containment measures have decreased the average contacts rapidly to a very low value. This value has been maintained until assumptions of lifting the containment were made and a sensitivity analysis was performed to indicate the range of results corresponding to different options and levels of lifting. In keeping with Tang and colleagues [23], the contact rate $c\left(t\right)$ is a decreasing function with respect to time t, which is given by
$$c_{1}\left(t\right)=\left(c_{0}-c_{b}\right)e^{-r_{1}t}+c_{b}$$
where $c_{0}\;$is the contact rate at the initial time (=14.781 according to Tang [22]), $c_{b}$ is the minimum contact rate under the current control strategies in Italy, and $r_{1}$ is the exponential decreasing rate of the contact rate. Then a contact releasing function was also defined in terms of the release time $T_{c}$, namely the time when lifting of the containment starts (after 60, 90 and 120 days in our sensitivity analysis (see Section 3). The exponential time scales are given by the constant $r_{1}$ and $r_{11}$ set at 1.3 and 0.5 $days^{-1}$, respectively.
$$c_{2}\left(t\right)=\left(c_{f}-c_{b}\right)(1-e^{-r_{11}\left(t-T_{c}\right)})+c_{b}$$
so that the total contact function is
$$c\left(t\right)=\Theta \left(t-T_{c}\right)c_{1}\left(t\right)+\Theta \left(T_{c}-t\right)c_{2}\left(t\right)$$
where $\Theta \left(t\right)$—not to be mistaken with θ (Table 1)—is a step function equal to 1 for t < 0 and 0 otherwise. Similarly, we set the diagnosis rate $\delta_{I}\left(t\right)$ for symptomatic infected individuals to be an increasing function with respect to time t using a slightly modified formula by Tang and colleagues [23]:$$\delta_{I}\left(t\right)=(\delta_{I0}-\delta_{If})e^{-r_{2}t}+\delta_{If}$$
where $\delta_{I0}=\delta_{I}\left(0\right)$ is the initial value of the diagnosis rate and $\delta_{If}=\underset{t\to \infty }{\lim}\delta_{I}\left(t\right)$ is the maximum (final) diagnosis rate with $\delta_{I0}<\delta_{If}$. This assumption provides a measure of the available resources to face the pandemic.

As data became available, it became apparent that the most reliable data in our possession were the number of hospitalizations. It is difficult in the development of the emergency to get reliable data on infected and symptomatic, and even the number of deceased people is subject to changing classification or failure to classify them accordingly. Therefore, it appeared that the best way to constrain the model was to rely on the number of hospitalizations included in compartment H. We used a simple nudging technique to constrain the model to reproduce the evolution of the hospitalizations during the development of the event [25,26]. The nudging was introduced in the equation for H by adding a term
$$-\tau \left(H\left(t\right)-H_{Obs}\left(t\right)\right)$$
where τ is the nudging time in inverse days—the shorter the time, the stronger the constraint. The results are shown in Figure 2 for Lombardy. These represent ensemble experiments obtained varying the minimum contacts value reached by the containment measures, which is obviously difficult to measure with certainty and therefore is a suitable candidate for sensitivity. The values change from one contact per day, obviously a very strict confinement, to 3–4 contacts, still much less than the pre-incident average contacts estimated at around 15 contacts per day for dense settlements situations. The observed hospitalizations are well within the envelope of the ensemble, so the model is capable of giving information on the worst- and best-case development.

The model can now be used to predict the evolution of the spread of the infection and its burden on hospitalizations under different conditions of social distancing.

## 3. Results

Figure 3 shows the results for Lombardy using nudging until the data available up to 12 April. The picture shows the envelope for simulations obtained varying the maximum confinement parameter $c_{b}$ from 1.0 to 2.4 contacts/day and then lifting it to a final value $c_{f}$. In all cases, the final contacts achieved have been set to 3.0. The overall dynamic shows that a strict containment ($c_{b}$ = 1.0) is capable of reducing severely the outbreak, but for any larger value there is significant tail of cases into the summer.

The amount of hospitalizations is depending on the sustained contacts value $c_{f}$ quite significantly, but higher values are also sensitive to the duration of strict containment measures (Figure 4). The duration of the confinement measures in Lombardy was set to a duration of 60 days from 9 March 2020 (Figure 4a,b)—we are showing here the consequences of extending the period to 90 and 120 days (Figure 4c,d). Because of the large size of the epidemic in its early stages, it is required to maintain the number of daily contacts still to a very low value. The value can be increased if the containment measures are extended over a longer period (Figure 4c,d), because the suppression of the infection is more effective and therefore higher values of contacts are sustainable.

The situation in Emilia-Romagna is different (Figure 5). In this case, the size of the epidemic is lower and, consequently, the containment measures damp the amount of hospitalizations to a sustainable number even when the final contact rate is 7.4, which is only half of the pre-incident value.

### Sensitivity Analysis on Initial Conditions and Parameters

The results of this analysis are presented in the Appendix A. If we hypothesize that on 9 March there were 5 undetected cases to 1 detected case instead of 10 to 1, the extended SEIR model predicts a lower number of hospitalizations in the summer for both Lombardy and Emilia-Romagna—this should not surprise, because the size of epidemic is now assumed to be smaller in its earlier stages. On the contrary, the initial proportion of asymptomatic cases has virtually no effect on our forecasts. Lastly, we reran all analyses by varying the initial value of the contact rate, because $c_{0}$ reflects social structures and habits that are likely to be different in Italy and China (e.g., family interactions, use of facemasks, etc.). Interestingly, we found that our results are very robust when the assumption $c_{0}\approx 15$ is violated.

## 4. Discussion

In this work, we used a forecasting method based on the number of COVID-19-associated hospitalizations, which is currently the most reliable information at our disposal, as well as the main indicator to predict the impact of the epidemic on the health services. These estimates are of great importance to make decisions and develop targeted strategies during the epidemic.

Our results indicate that the best parameter to assess the effectiveness of confinement measures and the risk of uncontrolled diffusion of the infection is the average number of daily contacts in a population (c). Though it is obviously not easy to come up with a totally objective way of monitoring c in a social setting, it is still easier to conceive ways of doing that rather than the more sophisticated $R_{0}$ index of morbidity, which includes individual-specific response to the virus and to circumstances of the infection. In a general social sense, if we measure c with respect to our pre-incident situation it is possible to assess heuristically that we should cut daily contacts by half in Emilia-Romagna (c=7.4) and by more than two thirds in Lombardy (c=3) to contain the spread of COVID-19. It might be difficult to translate this evidence into actual policy recommendations; however, the usage of geographically located data from personal devices may provide a quantifiable, reproducible and maybe predictable measure of daily contacts for communities and regions without infringing on privacy issues. Such measures could be the basis for informing appropriate policies during and after the incident. Our results also suggest that Lombardy is extremely sensitive to the number of daily contacts, that is, if $c_{f}$ increased up to 3.5, restrictive containment measures should be necessarily prolonged at least until early July to avoid a resurgence of hospitalizations.

Another point that emerges from our analysis is that Italy’s regional health systems can tolerate different levels of social contacts and still keep the infection rate within the capacity of their healthcare services. Because we have specified the same parameter values for the two study regions, such differences have to be found in the different initial conditions that have pushed Lombardy and Emilia-Romagna in different states from which the epidemic has then evolved.

On 20 February 2020, a case of COVID-19 was identified in Codogno, Lombardy, and in the next 24 h the number of reported positive cases increased to 36; it was immediately clear that there was a cluster of unknown size and that additional spread was probable [4]. Considering the number of cases and the advanced stage of the disease, it has been hypothesized that the virus was circulating in the population since January [5]. The outbreak rapidly evolved, with an increasing number of cases reported across the whole country, but with a marked involvement of Lombardy and more generally northern Italy, including Emilia-Romagna [5]. Lombardy epidemic was a few days ahead of the rest of Italy, and this might have to do with its strong productive structure that led to a rapid spread of the virus in some industrial areas [27]. As already said, the difference in the number of contacts to slow the spread is strongly linked to the extent of the early phase of the outbreak in the two regions. However, it should be recognized that Lombardy and Emilia-Romagna had different initial approaches to face the emergency, which reflect the different organization of their regional health systems. On one hand, Lombardy’s strong hospital system, coupled with a less strong territorial system [28], might have created greater stress on its hospital care services [27]. On the other hand, the strong system of public, territorial, and community welfare of Emilia-Romagna [29] adopted a mixed approach, based on both hospital care and territorial care [30]. Indeed, we found that on 9 March 2020 the persons under home isolation in Lombardy and Emilia-Romagna (L in our extended SEIR model) were 26% and 46% of all positive cases, respectively [6].

In this work, we implemented a deterministic equation-based model derived from the SEIR one. We chose this approach because it allowed including most of the data made available by the Civil Protection Department of Italy at the time of our modelling and computation. As more details will be made available, the model could be significantly improved. For example, including information about age structure could result in a new formulation of the system of differential equations to distinguish compartments of people in different age groups, even in the same state of the disease. This could also lead to new estimates for the number of daily contacts c depending on the different age groups considered.

Going beyond the purposes of this study, a mixed approach to modelling that combines equation- and agent-based methods could be adopted. This would need accurate and very stratified data and could be realized considering population subgroups, for example in the context of a neighborhood or a city. In this case, one could experiment with changes among different patterns of interactions depending on political and administrative decisions. This would have the potential to make rather accurate “what-if” experiments and provide more operational indications to policymakers about the “phase two” of coexistence with the virus.

## 5. Conclusions

Analyzing the burden of hospitalizations under different conditions of social distancing allows foreseeing the impact of the coronavirus pandemic on health services. This is of crucial importance for policy makers when a gradual lifting of containment measures needs to be planned.

## Figures and Tables

**Figure 1 jcm-09-01492-f001:**
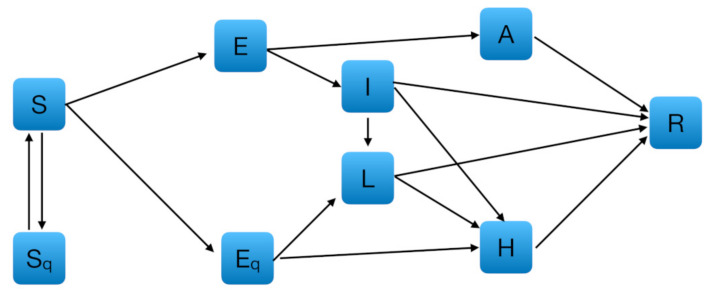
Diagram of the extended Susceptible-Exposed-Infectious-Recovered (SEIR) model adopted for simulating the spread of coronavirus disease 2019 (COVID-19) in Lombardy and Emilia-Romagna. S: susceptible, $S_{q}$: quarantined susceptible, E: exposed, $E_{q}$: quarantined exposed, I: infectious with symptoms, L: isolated infectious, A: infectious without symptoms, H: hospitalized, R: recovered.

**Figure 2 jcm-09-01492-f002:**
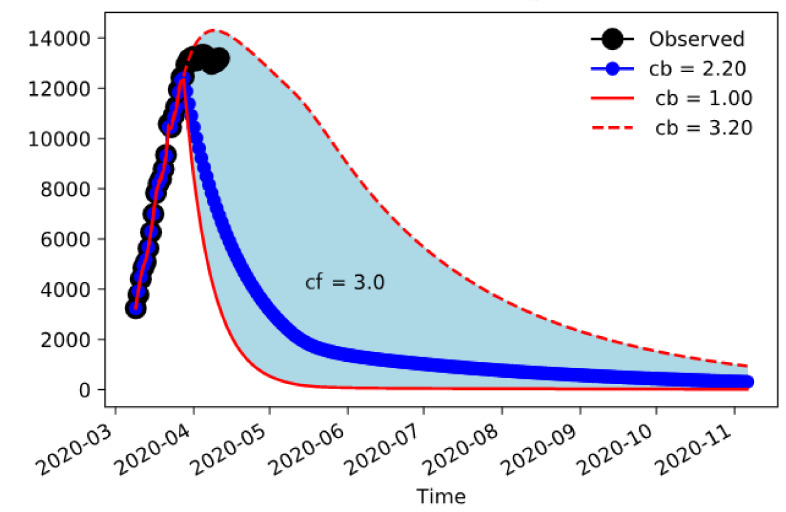
Number of COVID-19-associated hospitalizations in Lombardy according to the extended SEIR model. The black dots are observations, the red lines are the extrema of the ensemble, and the blue line is the center value. The envelope of the ensemble was calculated varying the minimum number of contacts reached by the containment measures from 1.0 to 3.2 in steps of 0.2. Nudging was performed until 15 days before 12 April.

**Figure 3 jcm-09-01492-f003:**
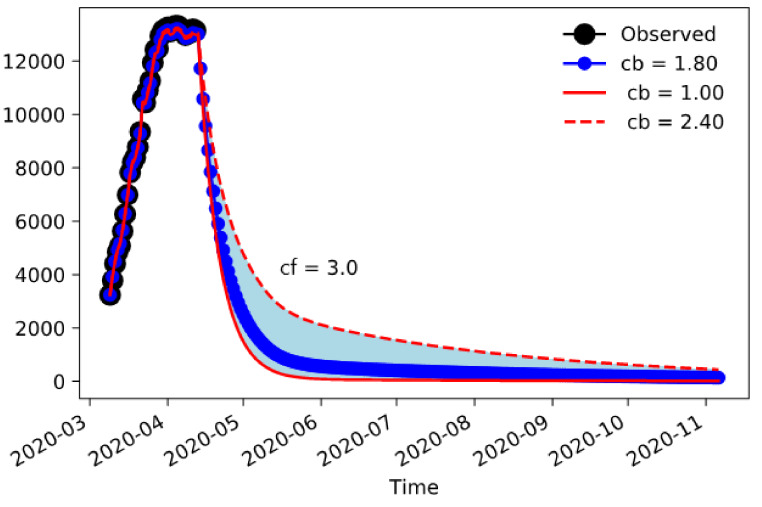
Lombardy’s COVID-19-associated hospitalizations with nudging between 9 March and 12 April, and forecasting until November 2020.

**Figure 4 jcm-09-01492-f004:**
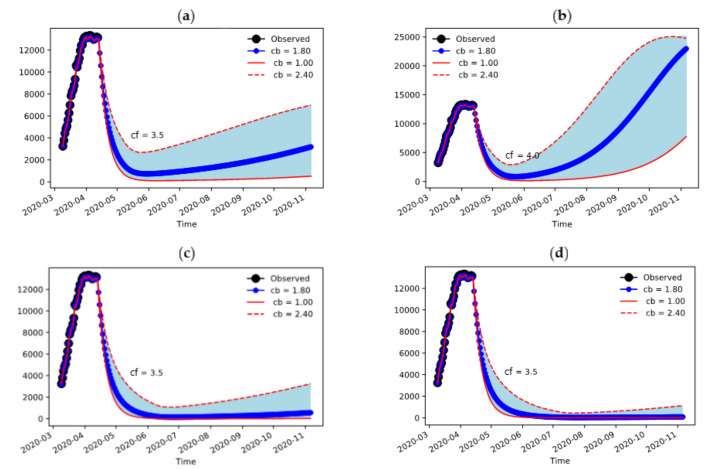
Sensitivity to final confinement value and duration of containment measures: (**a**) Final value of 3.5 contacts/day and containment lasting 60 days; (**b**) Final value of 4.0 contacts/day and containment lasting 60 days; (**c**) Final value of 3.5 contacts/day and containment lasting 90 days; (**d**) Final value of 3.5 contacts/day and containment lasting 120 days.

**Figure 5 jcm-09-01492-f005:**
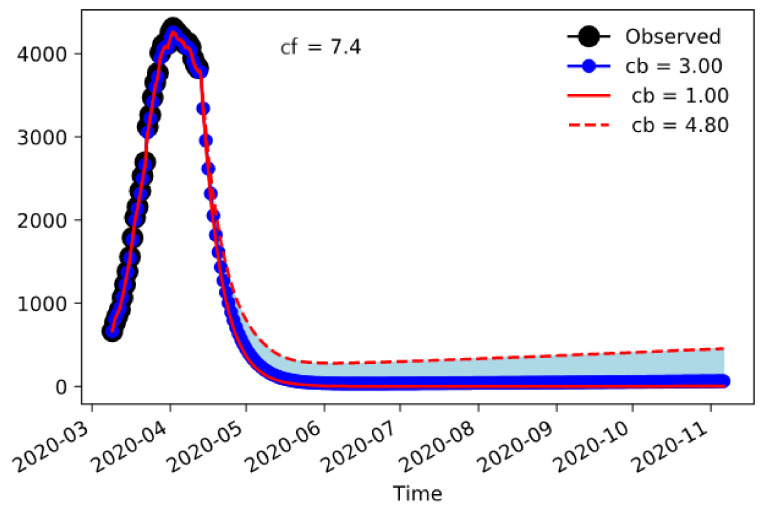
Emilia-Romagna’s COVID-19-associated hospitalizations with nudging between 9 March and 12 April, and forecasting until November 2020.

**Table 1 jcm-09-01492-t001:** Parameters for the extended Susceptible-Exposed-Infectious-Recovered (SEIR) model, Lombardy and Emilia-Romagna.

Parameter	Value	Definition
β	2.1011 × 10^–8^	Probability of transmission per contact
q	1.8887 × 10^–7^	Quarantined rate of exposed individuals
σ	1/7	Transition rate of exposed individuals to the infected class
λ	1/14	Rate at which the quarantined uninfected contacts are released into the wider community
ρ	0.86834	Probability of having symptoms among infected individuals
$\delta_{q}$	0.1259	Transition rate of quarantined exposed individuals to the hospitalized infected class
$\gamma_{I}$	0.33029	Recovery rate of symptomatic infected individuals
$\gamma_{A}$	0.13978	Recovery rate of asymptomatic infected individuals
$\gamma_{H}$	0.11624	Recovery rate of hospitalized infected individuals
α	1.7826 × 10^–5^	Disease induced death rate
θ	0.05	Infected rate of asymptomatic/symptomatic
$\varepsilon_{I}$	0.2000	Rate of home isolation for infected individuals
$\varepsilon_{q}$	0.2000	Rate of home isolation for quarantined exposed individuals
$\gamma_{L}$	0.13978	Recovery rate for isolated infected individuals
$\delta_{L}$	0.2000	Hospitalization rate for isolated infected individuals

**Table 2 jcm-09-01492-t002:** Population sizes initialized in the extended SEIR model, Lombardy and Emilia-Romagna, 9 March 2020. The compartments included in the model are in boldface.

Definition	Prevalent Cases		Source/Calculation
	Lombardy	Emilia-Romagna	
Resident on 31 October 2019 (*P*)	10,085,021	4,468,023	Istat estimate
Deaths (*D*)	333	70	Civil protection
Hospitalized (***H***)	3242	666	Civil protection
Isolated infected (***L***)	1248	620	Civil protection
Known infected (*H + L + D*)	4823	1356	Civil protection
Undetected infected (*A + I*)	48,230	13,560	(*H* + *L* + *D*) × 10 ^a^
Undetected asymptomatic infected (***A***)	32,153	9040	(*A* + *I*) × 2/3 ^b^
Undetected symptomatic infected (***I***)	16,077	4520	(*A* + *I*) × 1/3
Tests (*T*)	20,135	4906	Civil protection
Quarantined (*Q*)	15,312	3550	*T* − (*H* + *L*+ *D*)
Quarantined exposed (***E_q_***)	24	6	*Q* × 0.0016 ^c^
Quarantined susceptible (***S_q_***)	15,288	3544	*Q* × 0.9984 ^c^
Unknown exposed (***E***)	2212	513	*E_q_* × 90.277 ^c^
Recovered (***R***)	646	30	Civil protection
Susceptible (***S***)	10,013,798	4,449,014	*P – Q – E – H – L –* *D – A – I – R*

^a^ Assuming a ratio of 10 to 1. ^b^ Assuming that about two-thirds of the infected are asymptomatic. ^c^ See Tang et al. [22].

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
