# Peer review of "Forecasting COVID-19-Associated Hospitalizations under Different Levels of Social Distancing in Lombardy and Emilia-Romagna, Northern Italy: Results from an Extended SEIR Compartmental Model"

_jcm, 2020, doi:10.3390/jcm9051492_

Round 1

Reviewer 1 Report

I have a few suggestions for your consideration.   The line number is listed with the comment following. 

66.  should be  “create”

88-89.  I would recommend that these sentences are really two halves of the same thought and should be divided perhaps with a semi-colon as opposed to a period.

93. I would remove “for the first time” and instead say “Recovered) model, developed by Kermack and McKendrick(17).

104-106.  I am unsure what you are trying to say with this sentence.  Specifically on line 104 the component that says “we have to precise that also models”  I do not understand what you are trying to communicate there.

110. perhaps include “published, and become more and more refined”

Reviewer 2 Report

This manuscript describes a compartmental ODE model for COVID-19 in 2 regions of northern Italy, used to forecast hospitalizations under different containment approaches.

My biggest concern is the lack of sensitivity analysis. Some parameters are assumed or estimated with much uncertainty. A global sensitivity analysis would be best, but at least a local sensitivity analysis of the assumptions in the footnotes of Table 2 would help.

I am also concerned that the model predicts a steep decline in hospitalizations immediately after the nudging period in Lombardy. Has this been observed at this point?

Specific comments:

Lines 81-87 and 94-108 are not really needed. This manuscript does not use or compare to ABMs or statistical models, so more detail on those is not needed.

140: the parameters c and q are not defined in Table 1 or in the text below the ODE – please add them to one or both.

166: please state the assumptions made

Reviewer 3 Report

Referee Report JCM-792622

Forecasting COVID-19-associated hospitalizations under different levels of social distancing in Lombardy and Emilia-Romagna, northern Italy: results from an extended SEIR compartmental model

by Chiara Reno, Jacopo Lenzi, Antonio Navarra, Eleonora Barelli, Davide Gori, Alessandro Lanza, Riccardo Valentini, Biao Tang, Maria Pia Fantini

This paper uses essentially the same extended SEIR model proposed by Tang et. al. (JCM, February 2020) except that it includes a new class of isolated symptomatic individuals, L. However, focus and methods in this paper are quite different from Tang et. al.’s My general impression is that the authors made too many assumptions not based on data, especially the contact function c(t) (line 191), while Tang et. al. assumed that c = 14.781. Here some of my other comments.

1. What is the difference between the symptomatic class I and the isolated symptomatic class L? I think I and L are the same unless you assume that there are symptomatic people that are still running around in the city.
2. θ is used both as a constant (= 0.05) in the model equations and as a function in the definition of c(t).
3. There is a typo in line 148, it should be γAA.
4. In the paper, the authors define Tc to be the length of time social distancing is imposed, Tc = 30,60,90,120 days.

c1(t) = (c0 −cb)e−1.3t + cb

c2(t) = (cf −cb)(1−e−0.5(t−Tc)) + cb

            c1(t)   t < Tc

c(t) =  

             < c2(t)  t > Tc .

I wonder whether the results in Figure 4 depend on the above assumption of c(t).
5. Assumption was made that 1 - 0.86834 of the exposed class E becomes asymptomatic A. The model parameters in Table 1 are based on Tang et. al.’s paper. But that paper was applied to situation in Wuhan, China. Can we still use the same values for Italy? The two countries have different health care system and cultures. For example, people in China wear face mask but it is uncommon in Italy.
6. It would be interesting to plot the graphs of S(t),I(t) and H(t) from the simulations. Do they agree in ballpark with actual data?
7. From Table 2, S/Q = 0.00152 initially. Do we really need Sq and Eq? It is not possible to do contact tracing if you have tens of thousands of infected individuals.
8. The authors did not say anything about simulations of the model equations. How do they know their results are correct? There is also no sensitivity analysis on the model parameters, which is pretty common in this type of work.

Round 2

Reviewer 2 Report

All corrections have been made. I would still have liked a global sensitivity analysis of parameters, but the testing of the most uncertain parameters is a good choice.

Reviewer 3 Report

In the newly added supplementary material, I am confused about the captions. What does it mean "asymptotic cases are one-third of the infected pool at t_0"? t_0 is not given. I don't know the authors enforced that in the simulations. Also "undetected to detected ratio of 5 to 1." Again, what does that mean. Please refer back to the notations you use in the paper, like A, I, theta, etc. 
